# Geographic Variations in Dietary Patterns and Their Associations with Overweight/Obesity and Hypertension in China: Findings from China Nutrition and Health Surveillance (2015–2017)

**DOI:** 10.3390/nu14193949

**Published:** 2022-09-23

**Authors:** Rongping Zhao, Liyun Zhao, Xiang Gao, Fan Yang, Yuxiang Yang, Hongyun Fang, Lahong Ju, Xiaoli Xu, Qiya Guo, Shujuan Li, Xue Cheng, Shuya Cai, Dongmei Yu, Gangqiang Ding

**Affiliations:** 1National Institute for Nutrition and Health, Chinese Center for Disease Control and Prevention, Beijing 100050, China; 2Department of Nutritional Sciences, The Pennsylvania State University, State College, PA 16802, USA; 3National Cancer Center/National Clinical Research Center for Cancer/Cancer Hospital, Chinese Academy of Medical Sciences and Peking Union Medical College, Beijing 100050, China

**Keywords:** diet, dietary intake, spatial analysis, obesity, hypertension

## Abstract

Regional dietetic cultures were indicated in China, but how dietary patterns geographically varied across China is unknown. Few studies systematically investigated the association of dietary patterns with overweight/obesity and hypertension and the potential mechanism with a national sample. This study included 34,040 adults aged 45 years and older from China Nutrition and Health Surveillance (2015–2017), who had complete outcome data, reliable intakes of calorie and cooking oils, unchanged diet habits, and without diagnosed cancer or cardiovascular disease. Outcomes were overweight/obesity and hypertension. By using the Gaussian finite mixture models, four dietary patterns were identified—common rice-based dietary pattern (CRB), prudent diversified dietary pattern (PD), northern wheat-based dietary pattern (NWB), and southern rice-based dietary pattern (SRB). Geographic variations in dietary patterns were depicted by age–sex standardized proportions of each pattern across 31 provinces in China. We assessed the association of these dietary patterns with outcomes and calculated the proportion mediated (PM) by overweight/obesity in the association of the dietary patterns with hypertension. Evident geographic disparities in dietary patterns across 31 provinces were observed. With CRB as reference group and covariates adjusted, the NWB had higher odds of being overweight/obese (odds ratio (OR) = 1.44, 95% confidence interval (CI): 1.36–1.52, *p* < 0.001) and hypertension (OR = 1.07, 95%CI: 1.01–1.14, *p* < 0.001, PM = 43.2%), while the SRB and the PD had lower odds of being overweight/obese (ORs = 0.84 and 0.92, 95%CIs: 0.79–0.89 and 0.85–0.99, *p* < 0.001 for both) and hypertension (ORs = 0.93 and 0.87, 95%CIs: 0.87–0.98 and 0.80–0.94, *p* = 0.038 for SRB and *p* < 0.001 for PD, PMs = 27.8% and 9.9%). The highest risk of overweight/obesity in the NWB presented in relatively higher carbohydrate intake (about 60% of energy) and relatively low fat intake (about 20% of energy). The different trends in the association of protein intake with overweight/obesity among dietary patterns were related to differences in animal food sources. In conclusion, the geographic distribution disparities of dietary patterns illustrate the existence of external environment factors and underscore the need for geographic-targeted dietary actions. Optimization of the overall dietary pattern is the key to the management of overweight/obesity and hypertension in China, with the emphasis on reducing low-quality carbohydrate intake, particularly for people with the typical northern diet, and selection of animal foods, particularly for people with the typical southern diet.

## 1. Introduction

China has witnessed a remarkable nutrition transition over the past four decades, with hypertension and overweight/obesity becoming the leading risk factors for cardiovascular mortality and suboptimal diet among the top modifiable risk factor [1,2,3]. The temporal trends of the dietary transition and its impact on the prevalence of obesity and hypertension have been sustained and well documented [3,4,5,6]. However, little is known about the spatial variations in dietary patterns and their impact on overweight/obesity and hypertension, although the previous literature indicated the existence of regional dietetic cultures in China, and several national surveys have revealed similar regional aggregation phenomenon of cardiovascular disease and obesity [7,8,9,10,11].

China has a socioeconomically and topographically diverse landscape, both of which affect the dietary pattern and prevalence of cardiovascular diseases of local residents in different regions [7,11,12,13,14,15,16,17,18]. Previous studies involving the description of geographic characteristics of dietary patterns and their associations with overweight/obesity and hypertension in the Chinese population were simply based on regional food consumption features in common sense yet did not link the dietary patterns with the actual geographic distributions and either lack of national representativeness or of limited scope—they focused only on food consumption disparities and the different associations yet ignored potential mechanism, which makes the targeted public health effort on control of overweight/obesity and hypertension in both of national and subnational level lack fundamental knowledge [19,20,21,22,23,24]. By using nationally representative data from China Nutrition and Health surveillance (CNHS) (2015–2017), we aimed to depict the geographic variations in dietary patterns with spatial analysis, and systematically determine the association between them and the possible mechanism, among Chinese aged 45 years and older who had undergone the entire process of nutrition transition starting from the 1970s [6,11,25]. Since dietary patterns may be a reflection of broader lifestyle and socioeconomic status beyond dietary intake, we also explored the socioeconomic and lifestyle factors related to dietary patterns, which can be used for developing precise health promotion strategies [13,26,27].

## 2. Materials and Methods

### 2.1. Study Population

The CNHS (2015–2017) was national surveillance periodically conducted by the Chinese Center for Disease Control and Prevention (CDC), Beijing, China. The present study used data from adult surveillance in 2015. Details about designs, sampling methods, data collection, and contents of the surveillance were described in previous studies [7,28,29]. Briefly, the CNHS (2015–2017) covered 298 survey sites across 31 provincial administrative units (hereafter referred to as provinces) in mainland China. A stratified, multi-stage, probability-based random sampling scheme was used to select eligible participants aged 18 years and older, living in the sample area for more than six months during the last 12 months. For this cross-sectional study, we selected adults aged 45 years and older, excluding 22,364 participants with no available outcome data (body weight, height, and blood pressure); abnormal intakes of energy (<500 or >4000 kcal per day) and cooking oils (>150 g per day); previously diagnosed cancer or cardiovascular diseases; and changed diet habits due to obesity, self-report diagnosed hypertension, or other self-reported diagnosed metabolic disease, during the past 12 months (see Appendix A for a flow diagram of the study sample).

### 2.2. Assessment of Dietary Intakes

Diet was assessed using 3 consecutive days (including 2 weekdays and 1 weekend) of 24 h dietary recalls in addition to weighing household cooking oil and condiments. For a detailed diet survey, see Appendix A. Each food item (including cooking oils) was coded according to Chinese Food Composition Tables (FCTs) and classified into 35 food groups, from which intakes were then calculated, and 31 food groups were used for dietary patterns analysis according to consumption rates (Appendix A) [30,31,32]. Nutrient intakes per day and percentage of energy (%E) provided by macronutrients were also calculated. Cooking oils were classified into four types: oils with a high ratio of monounsaturated fatty acids (MUFA), oils with a balanced ratio of MUFA and polyunsaturated fatty acids (PUFA), oils with a high ratio of PUFA, and fats with a high ratio of saturated fatty acids (SFA), since the consumption of cooking oils in China has evident geographic disparities (see Appendix A in File S4 for definitions) [33]. Additionally, the participants’ habitual intake of food groups (including alcohol intake) during the last 12 months was assessed by a quantitative food frequency questionnaire (FFQ). The food groups listed in FFQ were classified into their nearest one of the 31 food groups. The correlation between short-term diet assessed by 3 consecutive days of 24 h dietary recalls and long-term diet assessed by FFQ was assessed (Appendix A) [34].

### 2.3. Assessment of Dietary Patterns and Their Geographic Variations

We classified subjects into mutually exclusive groups according to the most probable dietary pattern identified by the Gaussian finite mixture model (FMM)—a latent variable model. The advantages of FMM include that it accounts for the within-class correlation of dietary intakes (conventional methods assumed no residual covariance within class) and classification uncertainty and provides a relatively objective method to ascertain cluster numbers by comparing the many different clustering criteria [35,36]. The final model was identified according to the Bayesian information criterion (BIC), and four clusters—four dietary patterns, were selected [37]. They were named the common rice-based dietary pattern (CRB), prudent diversified dietary pattern (PD), northern wheat-based dietary pattern (NWB), and southern rice-based dietary pattern (SRB), according to their characteristics of food group intakes and geographic variations, and accounted for 40.8, 11.2, 26.4, and 21.5% of the study population, respectively (Appendix A). Detailed characteristics of nutrients and food group intakes of four patterns were displayed in results and Appendix A. We compared the proportion of different dietary pattern subgroups reaching or exceeding the recommended intake of dietary components in the newest Chinese Dietary Guidelines (CDG) to further understand the major dietary problems of each pattern (see Appendix A) [38,39,40]. The robustness of the characteristics of four patterns was assessed by a split sample validation (Appendix A). In order to describe the geographic variations in dietary patterns, we calculated the age–sex standardized proportions of various dietary patterns in the study population by 31 provinces with the Qinling Mountain Huaihe River (QMHR) line as the boundary of northern and southern China, which was used in spatial analysis and displayed by maps [11]. Then, to describe the rural–urban variations in dietary patterns, we calculated age–sex standardized proportions of various dietary patterns, respectively, in rural and urban areas of 31 provinces. Rural–urban ratio differences in dietary patterns were calculated by subtracting the urban proportion from the rural proportion for each pattern, for each province, and displayed by choropleth maps. We further derived dietary patterns with conventional factor analysis, calculated the age–sex standardized factor scores by 31 provinces, and compared the differences in derived dietary patterns of two methods, as well depicted the geographical disparities and associations of new-derived dietary patterns with investigated outcomes (Appendix A).

### 2.4. Assessment of Overweight/Obesity and Hypertension

Overweight/obesity was diagnosed by body mass index (BMI), and hypertension was diagnosed by blood pressure and medication history. Body weight and height were measured in the morning fasting state by trained investigators. The participants’ seated blood pressure after 5 min of rest was measured 3 times at 1 min intervals by trained staff from the local CDC. BMI ≥ 24.0 kg/m^2^ was deemed as overweight/obese, and systolic blood pressure ≥ 140 mmHg, diastolic blood pressure ≥ 90 mmHg, or taking antihypertensive drugs in the last two weeks were diagnosed as hypertension [41,42].

### 2.5. Socioeconomic Status (SES), Lifestyle, and Health Information

Household income and family members were collected through household questionnaires by trained investigators. Sex, age, ethnic group (Han, Zhuang, Manchu, Hui, Miao, Uygur, Yi, Tujia, Mongolia, Korean, Tibetan, and other ethnic), marital status (coupled or uncoupled), occupation (agriculture, manufacture, service, others, unemployed), physical activity, drinking behavior (never, moderate, or excessive), smoking behavior (never, ever, or current smoker), and health information (family history of cardiovascular diseases and diabetes) were collect with an individual questionnaire. Physical activity was measured with Global Physical Activity Questionnaire (MET-h/week), and drinking behavior was collected by the aforementioned FFQ. Drinking more than 15 g of alcohol per day was deemed as an excessive drinker, a drinking behavior but less than 15 g per day was deemed as a moderate drinker, and drinking frequency equal to zero was deemed as a never drinker [38].

### 2.6. Statistical Analysis

Intakes of 31 food groups were calculated as grams per 1000 kcal plus one (removal of extraneous variation in the study population and zero values) and then log-transformed before the FMM model was fitted [34,43]. Comparisons of 31 food group intakes and 14 nutrient metrics between four patterns were performed by a Kruskal–Wallis test or ANOVA. A Chi-square test was used to compare the proportion of different dietary pattern subgroups reaching or exceeding the recommended intake of dietary components. Global and local Moran’s I were used to detect spatial autocorrelation and local heterogeneity of dietary patterns. Age–sex standardized prevalence of overweight/obesity and hypertension by province were calculated and shown with choropleth maps. Rural–urban ratio differences in dietary patterns by province were compared with Rao Scott Chi-Square tests. Weights were calculated for the estimation of population-weighted indicators. Methods to calculate weights were reported by a previous study [7]. The socioeconomic structure of the 2015 Chinese population estimated by the State Statistics Bureau was the basis for the post-stratification weights.

Generalized logit models were used to estimate the associations of socioeconomic and lifestyle factors with dietary patterns and the associations of dietary patterns with overweight/obesity and hypertension. BMI was alternatively adjusted in our models for its potential mediation effects on the relationship between dietary patterns and hypertension [24,44]. Mediation effects of overweight/obesity (BMI ≥ 24 kg/m^2^) were analyzed with regression-based methods by decomposing the total effects of dietary patterns on hypertension into natural direct and indirect effects, and the proportion of mediation through overweight/obesity was calculated accordingly [24,45]. In order to examine other potential effect modifiers, we conducted stratification analysis among subgroups, including sex, age, occupation, residence place, and physical activity level. The joint effects of dietary patterns and physical activity on overweight/obesity were further analyzed. In order to explain the mechanism by which dietary patterns exert their impacts, a restricted cubic spline logistic model was used to assess the association between macronutrients and overweight/obesity and hypertension in four dietary patterns.

Basic statistical analyses used SAS version 9.4 (SAS Institute, Inc., Cary, NC, USA). Spatial analysis was performed by ArcGIS Desktop version 10.7 (ESRI, Inc., Redlands, CA, USA). Choropleth maps and other statistical analyses were generated by R Foundation for Statistical Computing version 4.1.3 (R Core Team, Vienna, Austria) [46].

## 3. Results

### 3.1. Geographic Variations in Dietary Patterns

Figure 1 showed evident geographic distribution disparities of four dietary patterns between northern and southern China and between rural and urban areas (see Appendix A for detailed data and geographic division).

CRB was all over the country and more in the south of the QMHR line, the northeast three provinces, and several provinces of the northwest. The highest proportion was in Jilin and the lowest in Tibet, while the higher proportion was in rural areas of the aforementioned regions (except Liaoning, Guizhou, and Yunnan) and in urban areas of Gansu (all *p* values < 0.05). By comparison, SRB was predominately distributed in the south of the QMHR line with smaller yet inconsistent rural–urban ratio differences among provinces in this area. It appeared SRB was more distributed in the rural areas of Yunnan and Guizhou in the southwest; however, the differences were not significant (*p* > 0.05 for both). Conversely, NWB was predominately distributed in the north of the QMHR line, with the highest proportion in Henan and the lowest in Yunnan. Notably, this pattern was not only distributed in the less developed rural areas of the northwest and the east but also in relatively developed urban areas of the north and the northeast. PD was the only dietary pattern predominately distributed in urban areas throughout China, while Shanghai (the Yangtze River delta region) had the largest proportion and Henan the smallest.

Figure 2 showed positive spatial autocorrelations for dietary patterns except PD (global Moran I: −0.003, *p* = 0.673). SRB showed the strongest spatial autocorrelation (global Moran I: 0.520, *p* = 0.000). A cluster of the high proportion of CRB was identified in Heilongjiang in the northeast and Guangdong and Guangxi in the south; a cluster of the high proportion of PD around Zhejiang (the Yangtze river delta region); a cluster of the high proportion of NWB in areas along and to the north of the Yellow River; and a cluster of the high proportion of SRB in provinces to the south of the Yangtze River except several provinces in the Yangtze river delta region (Zhejiang, Shanghai).

### 3.2. Characteristics of Dietary Patterns

For the typical northern dietary pattern, NWB had the highest carbohydrate intake and carbohydrate-to-energy ratio but the lowest protein-to-energy ratio and fat-to-energy ratio, as Table 1 shown. The median carbohydrate-to-energy ratio of NWB surpassed the recommended value in Chinese Dietary Reference intakes (DRIs), namely, 50–65% [47]. Notably, participants with NWB consumed the highest proportion of high-quality carbohydrates, of which coarse grains and tubers contributed the most since this population consumed fewer vegetables, particularly starchy vegetables, and moderate fruit (Figure 3 and Appendix A). Moreover, participants with NWB consumed the least protein from animal sources (only 20.3%), of which red meat contributed the largest proportion (85% of total meats, table not shown). Additionally, this group consumed the least dark green vegetables and SFA and the most PUFA.

Conversely, the typical southern dietary pattern—SRB had the highest intake of vegetables (particularly dark green vegetables and starchy vegetables), red meat, fish, and poultry. Additionally, they had the highest proportion of SFA; MUFA, particularly animal-source MUFA; and animal protein, yet the lowest proportion of PUFA. Participants with NWB and SRB had a similar proportion of excess sodium intake.

By contrast, participants with PD had the lowest proportion of excessive sodium intake, lowest intake of staple grains yet balanced proportion of rice and wheat, the lowest carbohydrate-to-energy ratio, and the highest intake of oils (particularly high-MUFA oils) and MUFA (Figure 3 and Figure 4, and Table 1). PD had a little lower intake of vegetables (including dark green and starchy vegetables) and red meat than SRB, and a little lower intake of coarse grains and tubers than NWB, while it still had similar ratios of macronutrients to energy and proportion of animal protein with SRB, a similar proportion of high-quality carbohydrate with NWB, yet the higher proportion of PUFA and lower of SFA than SRB, implying diversified food sources of animal protein and high-quality carbohydrate and larger proportion of MUFA from plant sources than SRB. Therefore, PD was a relatively balanced diet. CRB was similar to SRB, except for the lower intake of processed meat, offal, dark green and starchy vegetables, and SFA, as well as the higher intake of other vegetables, wheat, and PUFA.

Overall, a considerable proportion of the study population consumed excessive red meat (47.1%), sodium (41.6%), and saturated fatty acids (20.5%), but only a small proportion consumed adequate dairy (0.6%), fruits (1.4%), coarse grains (3.7%), nuts (13.6%), eggs (20.4%), fish (28.8%), poultry (29.2%), and vegetables (37.7%), especially dark green vegetables (26.4%), implying a generally suboptimal diet for the study population (Figure 4). 

### 3.3. Socioeconomic and Lifestyle Factors Associated with Dietary Patterns

With the CRB as a reference group (which characteristics closest to the average level, Appendix A), the PD and the NWB both had higher odds with higher education level and urban residence, but the PD had higher odds with higher household income while the NWB lower (Appendix A). The SRB also had lower odds with higher education levels. Moreover, people with SRB had higher physical activity while NWB was lower. All *p* values for the above comparisons were significant.

### 3.4. Association of Dietary Patterns with Overweight/Obesity and Hypertension

Figure 5 displays the estimated overall and subgroup associations between dietary patterns and overweight/obesity and hypertension, with CRB as the reference group.

After adjustment for BMI and other covariates, NWB showed a strong positive association with overweight/obesity (OR = 1.44, 95%CI: 1.36–1.52, *p* < 0.001) and was consistent across various subgroups. The positive association of NWB with hypertension (OR = 1.07, 95%CI: 1.01–1.14, *p* < 0.001) in overall subjects was significant in subgroups of females, rural residents, and agriculture. Compared to NWB, SRB had completely converse associations with overweight/obesity (OR = 0.84, 95%CI: 0.79–0.89, *p* < 0.001) and hypertension (OR = 0.93, 95%CI: 0.87–0.98, *p =* 0.038), which were basically consistent among various subgroups with exception of specific occupation and age group. PD also showed negative associations with overweight/obesity (OR = 0.92, 95%CI: 0.85–0.99, *p* < 0.001) and hypertension (OR = 0.87, 95%CI: 0.80–0.94, *p* < 0.001), consistent in various subgroups despite the magnitude differences.

BMI affected the associations of dietary patterns with hypertension. The positive association of NWB with hypertension substantially decreased (OR from 1.16 to 1.07) after BMI was adjusted, and the negative associations of PD and SRB with hypertension were also attenuated. By decomposing the total associations into natural direct and indirect effects, the PM by overweight/obesity varied greatly between different dietary patterns (Table 2). Overweight/obesity mediated more than 40 percent of the positive effect of NWB on hypertension, while less than 10 percent of the negative effect of PD. The natural indirect effects of overweight/obesity by which SRB exerted on hypertension was negative, suggesting the lower risk of hypertension of SRB was partly (27.8%) due to the lower risk of overweight/obesity for SRB.

Due to the lower physical activity level observed in participants with NWB (Appendix A), we further analyzed the joint effect of dietary pattern and physical activity on the risk of overweight/obesity and hypertension and found the associations of dietary pattern on overweight/obesity were modified by physical activity (P_interaction_ < 0.001, see Appendix A) [48]. However, even with the same high physical activity level, participants with NWB still had the highest risk of overweight/obesity and hypertension than participants with other patterns (see Appendix A for detail).

### 3.5. Associations of Estimated Percentage Energy from Macronutrients with Overweight/Obesity and Hypertension in Various Dietary Patterns

Multivariable splines for macronutrients varied among different dietary patterns (Figure 6). For overweight/obesity, splines of NWB for carbohydrate, fat, and protein presented a nonlinear trend of rising first and falling later, and the peak risk presented when 60% of energy was from carbohydrate and 20% of energy from fat (mid estimate from the spline), which represents a typical northern diet of high carbohydrate and low fat in the present study. However, with the increase in high-quality carbohydrates, splines of NWB and CRB for overweight/obesity both presented nonlinear decreasing trends. In terms of hypertension, the spline of NWB for carbohydrates presented a U-shaped curve, with the lowest risk being around 50% (mid estimate from the spline), while for protein, the spline of NWB presented a linearly decreasing trend. Moreover, the spline of CRB for protein and hypertension also presented a linearly decreasing trend. In contrast, splines of PD and overweight/obesity presented an increasing trend with the increase in carbohydrates and a decreasing trend with the increase in fat. The spline of SRB for protein and overweight/obesity presented a nonlinear increasing trend. The divergence in splines trends for protein intake and overweight/obesity between the SRB and the other three patterns when percentage energy from protein exceeded 15% (mid estimate from the spline) was related to the higher intake of processed meat and offal in animal foods (Appendix A). Additionally, as the percentage of energy from carbohydrates continued to increase (>60%), the differences in dietary components between different dietary patterns gradually narrowed and disappeared.

There were no significant associations of percentage energy from carbohydrates and protein in SRB and PD with hypertension, nor significant associations of fat and high-quality carbohydrates with hypertension in any dietary patterns.

## 4. Discussion

With a national sample of Chinese adults, we found that dietary patterns in contemporary China had evident North-South and rural–urban distribution disparities, clear spatial clusters bounded by the Yellow River and the Yangtze River, certain socioeconomic and lifestyle characteristics, and different associations with overweight/obesity and hypertension. NWB had a higher risk of overweight/obesity and hypertension compared to SRB and PD, and quite a portion (43%) of the higher risk of hypertension in NWB was mediated by overweight/obesity. Physical activity had a significant modifying effect on the association of dietary patterns with overweight/obesity. Furthermore, the highest risk of overweight/obesity in NWB appeared when carbohydrate intake was relatively high (60% of energy) and fat intake relatively low (20% of energy), while the optimal percentage of energy from carbohydrates for people with NWB seemed to be around 50% for the least risk of overweight/obesity and hypertension. Additionally, a higher intake of processed meat and offal was associated with a higher risk of overweight/obesity in the SRB and a higher risk of hypertension in all participants.

The present study demonstrated the dietary problems during nutrition transition, including excess intake of sodium, oils, and SFAs; a high proportion of refined grains in total grains; and inadequacy of vegetables, fruits, whole grains, dairy, fish, etc., consistent with previous studies, indicating the need for optimization of dietary patterns for the whole population [49,50]. Meanwhile, some merits of the traditional diet were preserved in the present study, such as the relatively high intake of coarse grains and tubers in NWB and plentiful vegetables, fish, and poultry in SRB, which are consistent with previous studies [5,6,49,51].

The most interesting part of this study is that we found that the typical northern and southern dietary patterns spatially clustered in provinces along and to the north of the Yellow River and along and to the south of the Yangtze River, respectively, which indicated the existence of regional dietetic cultures and great influence of external environment and climate on the local diet even in contemporary China [10,11,52]. In ancient times, there were records of food culture differences between northern and southern China [10,11,52]. However, no study investigates the geographic variations in dietary patterns in transitional China with rapid urbanization [11,19,20,23,52,53,54]. The present findings confirmed the disparities of dietary patterns in geographic distribution with spatial analysis and provided more details about the differences in dietary components among the four dietary patterns, which can provide an empirical basis for targeted investigation and dietary intervention.

It is more interesting that people with a typical southern diet have lower education levels yet higher physical activity levels and higher intake of vegetables, poultry, and fish than people with the northern one, which indicates the great impact of external environmental factors besides individual factors that deal with this discrepancy in dietary and lifestyle choices between the two regions represented by the two typical diets since higher education level has been consistently associated with better health behaviors and dietary quality in previous studies [13,26,55,56]. The Lifelines cohort study in the Netherlands also indicated the existence of regional “food cultures” with spatial analysis, by which the author means not only physical food environments but also social and cultural correlates of dietary behavior [57]. In general, physical food environments such as available fresh markets and food production and delivery system were suggested to impact local food consumption [58,59,60]. Moreover, dietary customs have some influence on dietary behaviors [10,61]. Additionally, the fact that PD was predominantly distributed in urban areas and developed areas and adopted by people with higher education and income was an embodiment of disparities in dietary intakes between rural and urban areas and among different SES groups in China [5,16]. However, the present study does not address by which these geographic and SES disparities arise; we rather suggest the existence of external environmental factors (physical, cultural, and socioeconomic), besides the individual ones, that may be responsible for these geographic disparities of dietary patterns and their different associations with health outcomes. More in-depth studies on how these external environment factors impact one’s dietary and diet-related behaviors are needed, and these factors should be taken into consideration for guiding geographic-targeted and population-targeted public efforts and actions in China and other transitional countries.

Unexpectedly, PD spatially clustered in areas where the traditional Jiangnan diet is located; however, PD was different from the higher intake of wheat and processed meat [11,38,62]. We speculated that PD might represent a North–South fusion diet since people in developed areas, such as the Yangtze River Delta region and urban areas, might integrate diverse dietetic cultures, which in turn enriched local diets [10,63].

The present study revealed that quite a part (43%) of the effect of the typical northern diet on hypertension was the higher risk of overweight/obesity, while the predominant part was the direct effect of dietary patterns. A previous study showed that 60 to 75% of the risk for primary hypertension was caused by overweight/obesity, and higher BMI mediates quite a part of the association of diet with diabetes [8,64,65]. Few studies investigated the proportion of dietary patterns in diet-outcome relationships in the Chinese population, particularly with a national sample [24]. The present findings indicated that maintaining healthy body weight, or keeping energy balance, was an important part of the management of hypertension in provinces along and to the north of the Yellow River, while optimization of dietary pattern—dietary components combination was the key for management of overweight/obesity and hypertension at the national level, considering the popularity of suboptimal diet.

We found the relatively high carbohydrate intake (60% energy from carbohydrates) and the relatively low fat intake (20% energy from fat) might be associated with a higher risk of overweight/obesity in the NWB. The low-carbohydrate diet (<40% of energy) or low-fat diet (<20% of energy), which are better for long-term weight control (>12 months), were substantially investigated in randomized trials [66,67]. A six-month randomized controlled-feeding trial in the Chinese population demonstrated that a relatively higher intake of carbohydrates (66% of energy), close to the traditional diet, might be better to promote weight loss compared with a high-fat diet (40% of energy) [68,69]. Different from the restricted feeding condition in trials where participants were fed with a meticulously formulated diet, residents in free-living status may vary substantially in quantity and quality of dietary components and energy intake as well as energy expenditure. In this national sample of Chinese adults, more than half of them consumed more than 60% of energy from carbohydrates, of which 86% was from low-quality refined grains and starches; even, more than 66% of people with NWB ate more than 60% of energy from carbohydrate, and more than 60% of them ate less than 25% of energy from fat. In this occasion, without an energy limit, a dietary pattern characterized by relatively high carbohydrate intake, mainly from refined grains, and low intake of fat, protein, and vegetables, is very likely to have a high glycemic load, which induces excess energy intake, and thus promotes the fat deposition in the body [70,71]. Prospective studies on free-living residents are required to validate our findings, especially in transitional countries.

In agreement with a prospective study in Chinese adults, our finding of the U-shaped association between carbohydrate intake and hypertension in the NWB suggests that moderate carbohydrate intake (around 50% of energy, and thus slightly higher intake of fat and protein) can favor simultaneous control of overweight/obesity and hypertension even in the case of a high proportion of low-quality carbohydrate [72]. Furthermore, the nonlinear decreasing risk of overweight/obesity in the two patterns with the increase in high-quality carbohydrates, consistent with prior findings, indicates that it is low-quality carbohydrates that should be reduced on the occasion of the popularity of high carbohydrate intake in people with NWB [72,73,74].

The present study observed divergence in the trends of the association of protein intake with overweight/obesity among different dietary patterns with the increase in percentage energy from protein, particularly when it exceeded 15%. We deduced that this divergence was due to the different driving sources in animal foods for the increase in protein intake in different dietary patterns. The positive association of higher plant protein rather than animal protein with overweight/obesity in the overall study population did not apply to participants with SRB who had the highest intake of animal protein, processed meat, and offal. Evidence in dietary protein intake and overweight/obesity is inconsistent, and neither do animal or plant sources of protein [75,76,77]. A randomized controlled trial in the European population with habitual high intake of protein and animal protein indicated a lower weight regain in high protein intake groups after weight loss, while another trial in the US population with habitual high intake of protein and animal protein indicated no difference in weight maintenance among groups with isocaloric protein (animal or plant sources) or maltodextrin supplements [75,78]. Similar inconsistency presented in observational studies [76,79]. This issue warrants further investigation in China and other transitional countries with habitual low intake of animal foods. In contrast, the positive association of processed meat with overweight/obesity and hypertension was relatively confirmative [80,81,82,83]. Animal offal, such as liver and kidney, are rich in cholesterol and fat and are part of local cuisine in some regions. Previous prospective studies in the Chinese population indicated that fatty red meats were positively associated with abdominal obesity risk, and dietary pattern characterized by high offal intake was associated with higher cardiovascular risks [84,85]. Of note, we did not observe any association of red meat with overweight/obesity, no matter whether it was in the overall study population or each dietary pattern. A meta-analysis of prospective studies suggested a positive association between red meat with overweight/obesity and weight gain, but three of the four studies used to generate estimates were from western countries, which had a much higher red meat intake than Asian countries, and the remaining one in Chinese population suggested no association of lean red meat with overweight/obesity [81,86]. These findings, together with the present findings, indicated attention should be paid to the selection of meat types and reduction in processed meat intake, especially for those with the typical southern diet. In addition, the negative association of higher intake of dairy, fish, and poultry with hypertension, rather than processed meat, also indicated the importance of animal foods selection in dietary pattern optimization [87,88].

Our study had some limitations. First, dietary intake assessed by the 3 consecutive days of 24 h recall could not reflect the long-term diet. However, FFQ is more susceptible to measurement errors and incapable of capturing the detailed vegetable subgroups and some foods with geographic characters, such as chili and legumes, which were important for the present study. We assessed the correlation coefficients of major food group intakes between FFQ and the 3 consecutive days of 24 h recall and presented plausible results as previous study (Appendix A) [89]. Second, the cross-sectional study design limited the interpretation of the causal relationship between dietary patterns and overweight/obesity and hypertension. However, the possible inverse causal relationship in cross-sectional design was maximally reduced since we excluded participants who changed their diet for known cardiometabolic risks or diagnosed cardiovascular diseases. Third, after excluding some participants, the representativeness of the study population was affected, limiting the generalizability of the study results. This was inevitable. Fourth, self-reporting bias may affect the intake of food groups that are perceived to be healthy or unhealthy, which were not measured in this study [36]. Fifth, although BMI is a common measurement for overweight/obesity, body composition, especially the percentage of body fat, is another important measurement of obesity; the association of dietary patterns with different measurements of obesity was not extensively investigated in the present study. Lastly, as with any observational study, the observed associations might be partly due to the residual confounding for known confounding factors, such as income, education, physical activity, smoking, and drinking. We alternatively adjusted these factors and analyzed the modifying effect of them, yet the results were not materially altered.

## 5. Conclusions

In summary, the geographic distribution disparities of dietary patterns illustrate the existence of external environmental factors that might be responsible for the different combinations of dietary components and, thus, different associations with overweight/obesity and hypertension. Dietary pattern optimization is the key to managing overweight/obesity and hypertension in China, and the emphasis should be placed on reducing low-quality carbohydrate intake and increasing vegetable intake, particularly for people with a typical northern diet, and a selection of animal foods, particularly for people with a typical southern diet. Dietary guidelines and actions should have more geographic and cultural considerations besides socioeconomic factors in light of these findings. This section is not mandatory but can be added to the manuscript if the discussion is unusually long or complex.

## Figures and Tables

**Figure 1 nutrients-14-03949-f001:**
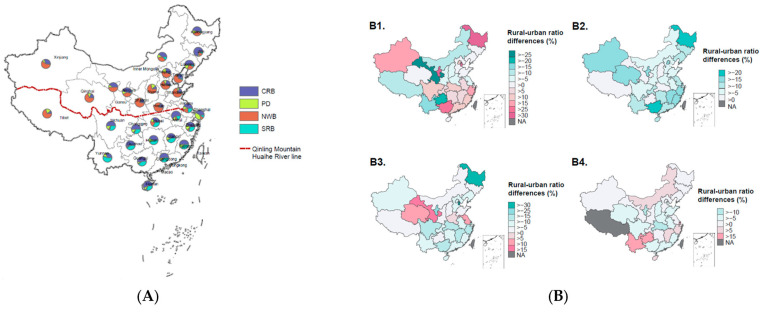
Geographic variations and rural–urban variations in dietary patterns across 31 provinces of China. (**A**). Age–sex standardized proportions of dietary patterns across 31 provinces of China; (**B**). The differences of age–sex standardized proportions of four dietary patterns between rural and urban areas; (**B1**–**B4**) represent common rice-based dietary pattern (CRB), prudent diversified dietary pattern (PD), northern wheat-based dietary pattern (NWB), and southern rice-based dietary pattern (SRB), respectively. CRB: common rice-based dietary pattern; PD: prudent diversified dietary pattern; NWB: northern wheat-based dietary pattern; SRB: southern rice-based dietary pattern; NA: not available.

**Figure 2 nutrients-14-03949-f002:**
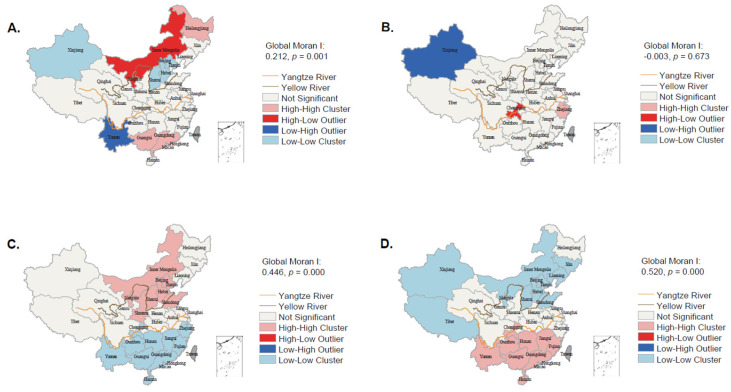
The clusters of provinces with high or low proportions for each dietary pattern; identified by spatial analyses. (**A**). Common rice-based dietary pattern; (**B**). Prudent dietary pattern; (**C**). Northern wheat-based pattern; (**D**). Southern rice-based pattern.

**Figure 3 nutrients-14-03949-f003:**
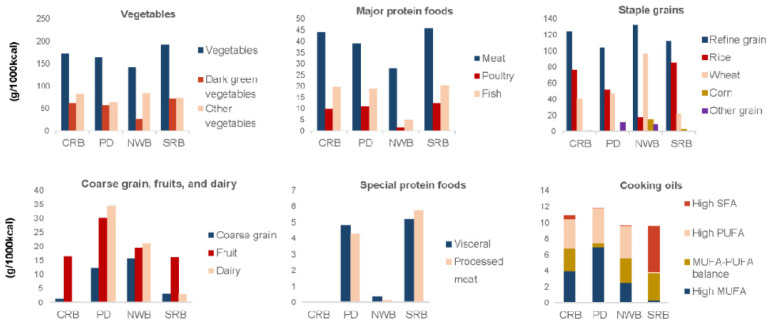
Mean intakes of selected food groups for four dietary patterns. Intake was calculated as grams per 1000 kcal energy. CRB: common rice-based dietary pattern; PD: prudent diversified dietary pattern; NWB: northern wheat-based dietary pattern; SRB: southern rice-based dietary pattern. Abbreviations: MUFA: monounsaturated fatty acids; PUFA: polyunsaturated fatty acids; SFA: saturated fatty acids.

**Figure 4 nutrients-14-03949-f004:**
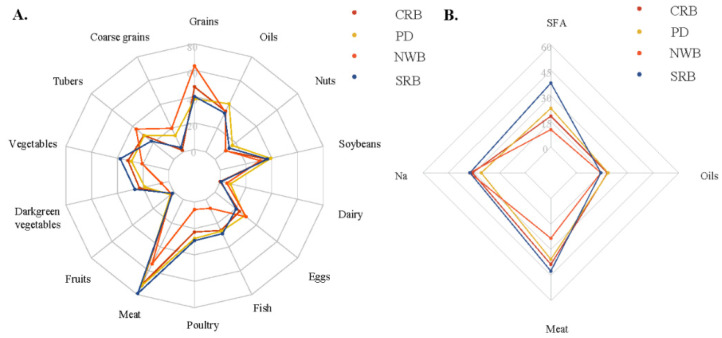
Proportion of subjects with different dietary patterns reaching or exceeding the recommended intake of dietary components. (**A**) Reaching the recommended intake; (**B**) Exceeding the recommended intake.

**Figure 5 nutrients-14-03949-f005:**
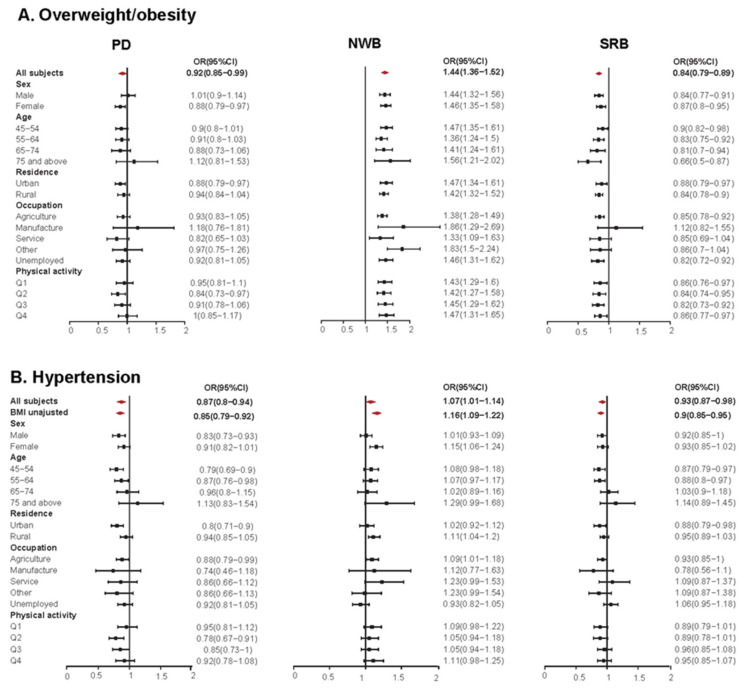
Estimated associations of dietary patterns with overweight/obesity, abdominal obesity, and hypertension among study population, with CRB as reference group. Analyses were adjusted for sex, age, ethnic group, education level, household income per capita, occupation, smoking behavior, drinking behavior, physical activity, total energy intake, and family history of cardiovascular disease and diabetes. The diamond and block represented the estimated odds ratios in overall and subgroup analyses, respectively. Q1–Q4: quartiles. CRB: Common rice-based dietary pattern; PD: Prudent diversified dietary pattern; NWB: Northern wheat-based dietary pattern; SRB: Southern rice-based dietary pattern; BMI: body mass index; OR: odds ratio; CI: confidence interval.

**Figure 6 nutrients-14-03949-f006:**
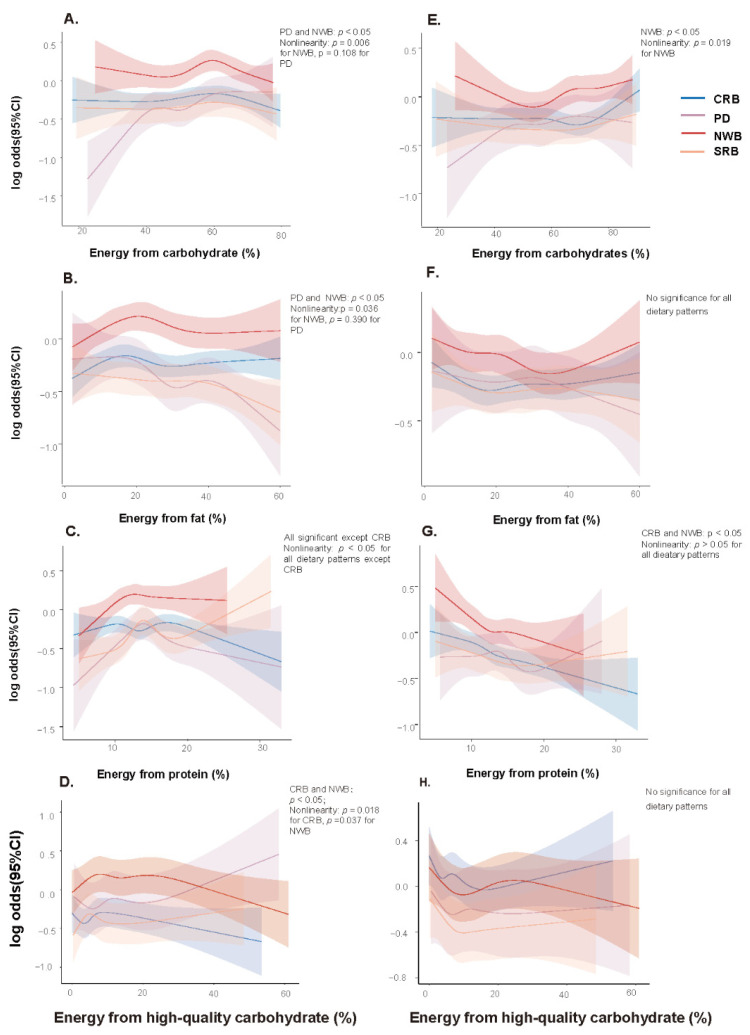
Associations between estimated percentage energy from macronutrients and overweight/obesity and hypertension stratified by dietary patterns. (**A**–**D**): overweight/obesity; (**E**–**H**): hypertension. All models were adjusted for sex, age, ethnic group, education level, household income per capita, occupation, smoking behavior, drinking behavior, physical activity, total energy intake, and family history of cardiovascular disease and diabetes. For hypertension, body mass index was adjusted in addition. CRB: Common rice-based dietary pattern; PD, prudent diversified dietary pattern; NWB, northern wheat-based dietary pattern; SRB, southern rice-based dietary pattern. For overweight/obesity, *p* _interaction_ < 0.01 for carbohydrate, fat and protein, and *p* _interaction_ = 0.015 for high-quality carbohydrate; for hypertension, *p* _interaction_ < 0.05 for percentage energy from carbohydrate and fat and *p* _interaction_ > 0.05 for protein and high-quality carbohydrate.

**Table 1 nutrients-14-03949-t001:** Nutrient intakes of study population with different dietary patterns, 2015 *.

Indicators	CRB	PD	NWB	SRB
Energy, kcal	1395 (1058,1801)	1530 (1220, 1927)	1495 (1171, 1899)	1540 (1165, 1978)
Protein, g	45.6 (33.8, 60.4)	53.2 (40.1, 68.9)	46.7 (36.0, 59.8)	52.1 (39.0, 67.5)
Protein, % kcal, mean (SD)	13.5 (4.0)	14.2 (3.9)	12.8 (2.8)	14.0 (4.0)
Animal protein, %	35.3 (19.9, 50.2)	41.7 (26.9, 54.5)	20.3 (8.4, 35.0)	42.3 (28.6, 55.0)
Total fat, g	40.6 (22.4,66.6)	50.4 (31.3, 75.6)	34.9 (18.1, 59.3)	46.4 (26.7, 74.2)
Total fat, % kcal	26.9 (16.8, 38.1)	30.6 (21.0, 39.7)	21.5 (12.7, 32.0)	27.8 (18.9, 38.3)
MUFA, % kcal	9.4 (5.5, 14.2)	10.7 (7.0, 15.5)	7.0 (3.7, 11.1)	10.4 (6.7, 15.0)
Animal MUFA, % kcal	4.6 (2.0, 7.8)	4.9 (2.6, 7.6)	2.3 (0.6, 4.6)	7.5 (4.4, 11.8)
PUFA, % kcal	4.3 (2.7, 8.9)	5.4 (2.9, 9.5)	4.4 (2.4, 8.9)	3.6 (2.4, 5.6)
SFA, % kcal	6.6 (4.3, 9.1)	7.5 (5.4, 9.7)	5.4 (3.4, 7.7)	8.4 (5.7, 11.6)
Total carbohydrate, g	196.4 (147.3, 261.4)	207.3 (160.6, 268.4)	240.5 (184.7, 304.7)	209.5 (156.0, 276.0)
Total carbohydrate, % kcal	59.3 (48.3, 69.8)	55.8 (46.3, 65.5)	66.6 (56.2, 75.5)	57.1 (46.7, 67.0)
High-quality carbohydrate, %	10.5 (6.3, 18.6)	18.7 (11.1, 30.1)	19.4 (10.6, 30.7)	13.9 (7.8, 23.0)
Fiber, g	6.8 (5.8, 18.2)	8.4 (5.9, 12.1)	9.5 (6.9, 12.9)	7.4 (5.4, 10.7)

* Data were described as median (interquartile range) unless otherwise specified. Kruskal–Wallis test or one-way analysis of variance was used to compare between-group differences (*p* < 0.0001 for all indicators). Abbreviations: CRB: common rice-based dietary pattern; PD: prudent diversified dietary pattern; NWB: northern wheat-based dietary pattern; SRB: southern rice-based dietary pattern; SD: standard deviation; MUFA: monounsaturated fatty acids; PUFA: polyunsaturated fatty acids; SFA: saturated fatty acids.

**Table 2 nutrients-14-03949-t002:** Mediation analysis of the associations between dietary patterns and hypertension mediated by overweight/obesity, with CRB as the reference group *.

Dietary Pattern	Mediator	OR^NIE^ (95%CI)	OR^NDE^ (95%CI)	OR^TE^ (95%CI) ^a^	PM (%) ^b^
PD	Overweight/obesity	0.98 (0.97–1.00)	0.86 (0.80–0.93)	0.85 (0.78–0.91)	9.9
NWB	Overweight/obesity	1.07 (1.06–1.08)	1.09 (1.03–1.15)	1.17 (1.10–1.23)	43.2
SRB	Overweight/obesity	0.97 (0.96–0.98)	0.92 (0.87–0.97)	0.89 (0.84–0.95)	27.8

* Abbreviations: OR^NIE^: odds ratio for the natural indirect effects; OR^NDE^: odds ratio for the natural direct effects; OR^TE^: odds ratio for the total effects; PM: proportion of mediation. Analyses were adjusted for sex, age, ethnic group, education level, household income per capita, occupation, smoking behavior, drinking behavior, physical activity, total energy intake, and family history of cardiovascular disease and diabetes. Uncertainty of estimations of exposure-outcome associations here used bootstrap-based method with 1000 replicates to obtain 95% confidence intervals (CIs). ^a.^ The estimated total effects of mediation analysis in this table were slightly different from that estimated in Figure 4 because the calculation of natural direct effects needs to be conditional on the level of the covariates. In this analysis, continuous covariates were fixed at their median and categorical covariates were set at levels observed most frequently in study population. ^b.^ Proportion of mediation = natural indirect effect/(natural direct effect + natural indirect effect) × 100.

## Data Availability

The data are not allowed to be disclosed according to the National Institute for Nutrition and Health, Chinese Center for Disease Control and Prevention.

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
