# Peer review of "Geographic Variations in Dietary Patterns and Their Associations with Overweight/Obesity and Hypertension in China: Findings from China Nutrition and Health Surveillance (2015–2017)"

_nutrients, 2022, doi:10.3390/nu14193949_

Round 1

Reviewer 1 Report

The manuscript titled “Geographic Variations of Dietary Patterns and Their Associations with Overweight/Obesity and Hypertension in China: Findings from China Nutrition and Health Surveillance (2015– 4 2017)” presents an assessment of dietary habits in a sampled Chinese population aiming at identification of dietary patterns different by province of residence and with specific relation with health outcomes including overweight/obesity and hypertension. The purpose is valuable, but several methodological concerns are present and must be clarified before the manuscript can be considered for publication.

The main concern is about the purpose of the study which is unclear. Authors should clarify the way they identified dietary pattern. Now, they stated that four patterns were selected, but the characteristics of each dietary pattern must be specified in order to clarify to the reader how they have been identified and how adherence of each participant to the different dietary patterns was measured. It seems that percentage of subjects divided by dietary habits are presented. However, dietary pattern generally are identified for specific characteristics and the level of adherence in a population may greatly vary. A table indicating the intakes of food used for the assessment and definition of the dietary patterns described in the study should be added.

The investigation of adherence to different dietary habits would allow also to investigate the relation with overweight/obesity, abdominal obesity and hypertension in all patterns (e.g. increasing tertiles or quartiles or nonlinear relation with spline analysis), without the need to use one pattern as reference, hampering the assessment of the relation of such pattern with the investigated outcome.

In addition, the description of dietary patterns will be helpful to interpret the analysis reported in Figure 6 with the % of energy from macronutrients, which can greatly vary depending of the type of diet and the intake of such macronutrients, e.g. animal proteins and high quality carbohydrates.

Without an analysis according to adherence, Authors can identify only that differences in dietary habits are present and are related to geographic location of provinces, as presented at L389-393, but a clear difference between patterns remains unclear to the reader and especially which suggestions/intervention from a public health perspective can be implemented to improve human health through changes in dietary habits. 

Although Authors provides some references about the representativeness of the sampled study population of the general Chinese population of the corresponding provinces, the presentation of data using choropleth maps can be considered misleading that the assessment was performed on the entire population. The correlation between adherence to dietary pattern based on the sampled population with sex and age-standardized obesity and hypertension is not clear. Authors reported that subjects with diagnosis of obesity, hypertension and other metabolic disease were excluded, but later these subjects seemed considered in the computation of obesity and hypertension. Despite the limitation of a cross-sectional analysis, Authors should assess the differences of dietary habits according presence/lack of hypertension and obesity. The exclusion of subjects who reported a change in dietary habits due to diagnosis of metabolic disease is a strength of such analysis. 

Minor notes:

L32: covariates could be reported for clarity or

L79: Study in uppercase

L92: please check Figure S1, the number of subjects excluded due to tumor and CVD are not reported in the box.

L104: It is not clear why ‘evident geographical disparities’ should be present for cooking oils. This seems a results of the analysis of the study and the identification of different dietary patterns within the study. Please explain and clarify this point.

L126-L129: Authors should clarify the differences of dietary patterns considered. In the Appendix 5, percentage of subjects are presented, but the detailed characteristics of food intake must be added, especially taking into account the journal topic.

L150-151: as per the exclusion criteria, please clarify which health information were eventually considered in the analysis. Some subjects should have been already excluded considering the flow-chart presented above. Please clarify.

Author Response

Dear reviewer,

Thanks for your patient evaluation to the manuscript and valuable suggestions. Here I uploaded the coverletter, which illustrated the details about the revisions and responses to your questions and suggestions.

May you a happy Mid-Autumn Festival and family reunion!

Sincerely,

Rongping Zhao, MD

NINH, Chinese CDC

Reviewer 2 Report

interesting research idea. please do the recommended revision specified as follows: 

research title: why consider both overweight and obesity?  

Abstract: use mesh standard for keywords. 

Line 19: Any fact available? every country has its own info regarding lifestyle, so on...

line21: more than 4 years past since your data collection? no publish up to now?

line 30: overweight/obesity: two different terms? separate their results...

line 36-37: wasn't it obvious prior to the study?

have you seen the following paper in your country done? if so, what is the major benefits of your study?

Rutayisire E, Wu X, Huang K, Tao S, Chen Y, Wang S, Tao F. Dietary patterns are not associated with overweight and obesity in a sample of 8900 Chinese preschool children from four cities. Journal of nutritional science. 2018;7.

Methode: 

The sampling method should be shown in a flowchart. 

line 113: include reference..

Figure 6: instead of using the description in the caption, have it in a separate sentence. 

Analysis:

in the abstract, no P-value has been reported, additionally, no Pvalie is found in your statistics report. also, F must be reported for Anova. 

Discussion: 

There can be as many intervening variables in such research which have a high potential to affect your study results. therefore, you should state them as study limitations 

*nutrition education affects eating patterns significantly, so, It should be measured as an effective factor and if not measured, must be stated as a study limitation. use the following reference to refer to the necessity of nutrition education: 

Makaracı Y, YücetaÅŸ Z, Devrilmez E, Soslu R, Devrilmez M, Akpınar S, Popovic S. Physical Activity and Nutrition Education Programs Changes Body Mass Index and Eating Habits of 12th Grade Students: An Intervention during the COVID-19 Pandemic. Annals of Applied Sport Science.:0-.

*another factor besides nutrition education is physical fitness which affects hypertension. probably,  Geographic Variations would affect the condition for physical activity (PA) and hence, affect your results..so, state that PA may be effective in the fitness of overweight and obese persons: 

you can use the following reference: 

Abdelkarim O, Ammar A, Trabelsi K, Cthourou H, Jekauc D, Irandoust K, Taheri M, Bös K, Woll A, Bragazzi NL, Hoekelmann A. Prevalence of underweight and overweight and its association with physical fitness in Egyptian schoolchildren. International journal of environmental research and public health. 2020 Jan;17(1):75.

*Although BMI is not an exact measurement for obesity or overweiht, its the easiest way for epediomo;pgical studies. However, state it as a limiting factor in youe study. suppose that one person is 170 cm, 80 kg, body fat percent: 20. in BMI, he is overwweight, but in PBF: normal. so, body composition is important. you can use the fillowing work for clarification:

Irandoust K, Taheri M, Mirmoezzi M, H’mida C, Chtourou H, Trabelsi K, Ammar A, Nikolaidis PT, Rosemann T, Knechtle B. The effect of aquatic exercise on postural mobility of healthy older adults with endomorphic somatotype. International Journal of Environmental Research and Public Health. 2019 Nov;16(22):4387.

Author Response

Dear reviewer,

Thanks for your patient evaluation to my manuscript and valuable suggestions. Here I uploaded the cover letter which illustrates the details of the revision and responses to your questions and suggestions.

May you a happy Mid-Autumn Festival!

Sincerely,

Rongping Zhao, MD

NINH, Chinese CDC

Reviewer 3 Report

Very great research topic. However, would the results be different if samples selected include population below 45-year-old? Different generations have different dietary habits. People above 45-year-old were born 1972 or before considering that data were collected in 2015-2017. Big changes have happened during that time frame in China, which might have affected people's dietary habit.

Author Response

Dear reviewer,

Thanks for your appreciation to the manuscript. Here I uploaded the coverletter which covered the details of the revisions and responses to your questions. Please refer to the "Responses to the comments in review report 3" for the responses to you.

May you a happy Mid-Autumn Festival and family reunion!

Sincerely,

Rongping Zhao, MD

NINH, Chinese CDC

Round 2

Reviewer 1 Report

In the revised manuscript, Authors clarified most of the previous concerns and adequately addressed the main issues.

Some further comments are below based on the revised version and Authors' replies:

The modality of data analysis and presentation not considering adherence to dietary habits is still of concerns. Despite the Authors explanation, I agree that the analysis classifying target population into mutually exclusive subgroups could be considered clearer to answer the second question. However, also distribution of adherence to the four dietary patterns in all provinces should depict the geographical differences. Still, data analysis by adherence to dietary patterns is suggested (if not as main results maybe as supplementary material).

In the exclusion criteria, I would specify in the main text too that "participants who changed their diet habits due to obesity, self-report diagnosed hypertension, or other self-reported diagnosed metabolic disease" were excluded. I see that this information is reported in the Appendix, but a clear indication in the main text is recommended.

Author Response

Dear reviewer:

Thanks for your careful review to this manuscript. We have added the dietary pattern adherence analyses to the supplementary materials according to your suggestions and make a clearer indication of the exclusion criterion of study participants. Please see the manuscript in the attachment.

Sincerely,

 Rongping Zhao

NINH, China CDC

Reviewer 2 Report

It seems that yo have just justifid your own report and havent ammended the manuscript based on recommendations. 

Author Response

Dear reviewer:

Thanks for your careful review to this manuscript.  According to your suggestions, we added the P-value reports in the abstract and the result, and amended the background introduction both in the abstract and main text. Besides, your valuable suggestions about other limiting factors of this study and the influence of physical fitness to the associations of dietary patterns with outcomes had been indeed amended in the first revisions. Please see the responses to the reviewer in review report 2 and the manuscript in the attachments.

Sincerely,

Rongping Zhao

NINH, China CDC
